# Investigating online psychological treatment for adolescents with a visible difference in the Dutch YP Face IT study: protocol of a randomised controlled trial

Marije van Dalen ![ORCID],[1] Suzanne G M. A Pasmans,[2] Marie-Louise Aendekerk,[1] Irene Mathijssen,[3] Maarten Koudstaal,[4] Reinier Timman,[5] Heidi Williamson,[6] Manon Hillegers,[1] Elisabeth M W. J Utens,[1,7,8] Jolanda Okkerse ![ORCID] [1]

For numbered affiliations see end of article.

**Correspondence to**
Dr Jolanda Okkerse;
j.okkerse@erasmusmc.nl

## ABSTRACT

**Introduction** This paper outlines the study protocol for the Dutch Young People (YP) Face IT Study. Adolescents with a visible difference (ie, disfigurement) often experience challenging social situations such as being stared at, receiving unwanted questions or being teased. As a consequence, some of these adolescents experience adverse psychosocial outcomes and appearance-related distress. To address this appearance-related distress, an online psychotherapeutic intervention, YP Face IT, has been developed. YP Face IT uses social interaction skills training and cognitive–behavioural therapy. The Dutch YP Face IT Study tests whether this intervention is effective in reducing social anxiety and improving body esteem.

**Methods and analysis** Participants are adolescents aged 12–18 with a visible difference and experiencing appearance-related distress. In this two-armed randomised controlled trial, 224 adolescents will be randomised to care as usual or YP Face IT. Adolescents will complete questionnaires at baseline, at 13 weeks and at 25 weeks. Primary outcomes are differences in social anxiety and body esteem between YP Face IT and care as usual. Secondary outcomes are differences in aspects of self-worth, perceived stigmatisation, health-related quality of life, life engagement, appearance-related distress and depressive symptoms between the two groups.

**Ethics and dissemination** Research ethics approval was obtained from the medical ethics review committee in Rotterdam (reference number MEC-2018-052/NL63955.078.18). Findings will be disseminated through academic peer-reviewed publications, conferences and newsletters to patient associations and participants of the study.

**Trial registration number** The Netherlands Trial Register (NL7626).

## Strengths and limitations of this study

► This is the first randomised controlled trial to assess Young People Face IT, an online psychosocial intervention for adolescents with a visible difference.

► Measurements include a direct follow-up and follow-up 3 months after completion of the intervention.

► Participants are recruited throughout the Netherlands, enhancing generalisability.

► Participants are not blinded to allocation, which could lead to reporting bias.

disfigurement) to the face or body occurs in 1 in 44 people, a visible difference solely to the facial area in 1 in 111.[1] A visible difference can be either congenital (eg, cleft lip and palate and craniofacial conditions), result from a skin condition (eg, vitiligo, psoriasis and acne), trauma (eg, burns and scars), disease (eg, cancer, meningitis and alopecia areata) or medical treatment (eg, surgery or radiotherapy). Whether someone identifies as having a visible difference is a subjective experience. A clear definition of what a visible difference is can thus not be given.

Some studies show that adolescents with a visible difference experience no additional psychosocial problems[2,3] and even perceive their friendship, social acceptance, appearance and emotional distress in a more positive way than unaffected peers.[4] However, reviews show that a significant number of adolescents with a visible difference do experience adverse psychosocial outcomes.[5–7] More specifically, they experience higher levels of minor psychological disturbance, self-consciousness in public situations, neuroticism,[8] internalising symptoms[7,9] and anxiety, depression and difficulties with social functioning.[10] Furthermore, parents perceive their children as being more shy and inhibited.[11]

## INTRODUCTION

Having a visible difference can have a substantial impact on one's life. People with a visible difference commonly encounter difficult social situations, such as being stared at, receiving unwanted questions and feeling isolated. A visible difference (ie,

Available interventions for adolescents or adults with a visible difference are scarce and lack evidence base.[12–15] Research often lacks a solid design and overall methodological quality and yields small effect sizes. However, there is some support for models involving social skills training (SST), cognitive–behavioural therapy (CBT)[12 14 15] and self-help for managing anxiety.[13]

Young People (YP) Face IT was developed in the UK, in collaboration with adolescents and the charity Changing Faces.[16] YP Face IT was derived from the adult intervention Face IT, which was found to be effective at reducing anxiety, depression and appearance concerns while increasing positive adjustment.[17] In an acceptability and feasibility study concerning YP Face IT, the intervention was found to be acceptable, with most of the adolescents thinking it would help them improve their confidence and self-acceptance and help them develop new skills in dealing with difficult social situations.[16]

## Aims

In sum, interventions aimed at adolescents with a visible difference are scarce and have limited evidence to support them. The available evidence, however, does point to SST and CBT as being potentially effective methods for managing psychosocial difficulties among people with a visible difference. In an attempt to provide adolescents coping with a visible difference with an effective intervention focussing on improving emotional resilience, the Dutch YP Face IT Study aims to test an online psychotherapeutic intervention combining social interaction skills training (SIST) and CBT. More specifically, the aims of the current study are to test the effectiveness of the Dutch version of YP Face IT in reducing anxiety and improving body-esteem among Dutch adolescents with a visible difference.

## METHODS AND ANALYSIS

To assess the effectiveness of the Dutch version of YP Face IT, a randomised controlled trial (RCT) will be conducted.

## Participants

This study will be conducted in a Dutch academic hospital. The target group for this study is Dutch adolescents aged 12–18 with a visible difference and experiencing appearance-related distress. Recruitment will take place throughout the Netherlands.

Adolescents will be eligible for participation if they are aged between 12 and 18, have a visible difference and access to a computer with internet and experience appearance-related distress. The appearance-related distress is operationalised as an elevated score on either social anxiety or depression or a lower score on body esteem, as assessed by self-reports.

Exclusion criteria are a mental disability, reading ability below 12 years of age, visual impairments preventing comprehension of the online intervention, a clinical diagnosis of depression, psychosis, body dysmorphic disorder or an eating disorder and insufficient proficiency of Dutch. Furthermore, adolescents who receive face-to-face care by a psychologist will be excluded from participation.

## Intervention

YP Face IT (Dutch: *Face IT voor jongeren*) is an online psychotherapeutic intervention[16] specifically designed for adolescents and developed in cooperation with adolescents and the British charity Changing Faces. The intervention is aimed at helping adolescents cope with some key concerns people with a visible difference may have, such as anxiety, depression, low self-esteem and appearance-related distress.

YP Face IT combines SIST with CBT. The intervention consists of 7 weekly sessions and an additional booster session after 6 weeks, helping the adolescent recall what was learnt during the weekly sessions. The first three sessions focus on SIST, sessions 4–6 focus on CBT and session seven is a summary of all learnt techniques. Session eight is a booster quiz, 6 weeks after session seven. The content of each session is displayed in table 1. Each session has different exercises and activities and takes 45–60 min to complete. In addition to the weekly sessions, participants will also be asked to complete small homework assignments between sessions. Participants will have access to an online journal in which they can write about their experiences and enter the answers to the homework assignments. Participants will have the opportunity to contact the research team if they are struggling with the homework activities. A reminder email is send if the participant has not entered data relating to the homework activity in their journal within 5 days following their last session.

YP Face IT is designed as a self-help intervention, without any active guidance by a psychologist. Adolescents work through the sessions on their own. Parents receive a short description of each session and will receive the reminders, but are not involved in the sessions themselves.

Monitoring will take place to assess whether adolescents complete the homework assignments and whether they are struggling in any way with the intervention or their mental health. Monitoring will be done by an experienced clinical psychologist and a master's student. Should we find that adolescents experience severe psychological symptoms, they are referred to other care that may better suit their needs.

## Care as usual (CAU)

CAU will consist of regular medical care. The type of the care may differ depending on the condition (eg, skin care for eczema and orthodontics for cleft palate). Psychological care is not part of CAU. However, medical personnel will monitor the adolescents' wellbeing as part of CAU. They may refer the adolescents to a psychologist in case of suspicions of psychological

van Dalen M, *et al. BMJ Open* 2021;**11**:e041449. doi:10.1136/bmjopen-2020-041449

**Table 1** Content of the YP Face IT intervention

| Session | Content | Type | Week |
|---------|---------|------|------|
| 1. Common problems | Common problems experienced by adolescents with a visible difference: bystander responses, teasing, bullying. | SIST | 1 |
| 2. Improve your social skills | Body language, verbal and non-verbal skills and building positive social skills. | SIST | 2 |
| 3. Don't be SCARED, REACH OUT | Effect of own behaviour on other people. Toolbox of techniques to cope with difficult social situations. | SIST | 3 |
| 4. Think, Feel, Do | Relationship between thoughts, feelings and behaviours. Identifying negative thoughts and using a positive voice to challenge these thoughts. Different coping strategies for negative thoughts. | CBT | 4 |
| 5. SMART goals | Using SMART goals to help achieve goals, dividing the end goal into subgoals. Dealing with concerns in romantic relationships. | CBT | 5 |
| 6. Beating anxiety | Exposure therapy in the form of a fear ladder. Relaxation techniques. | CBT | 6 |
| 7. Looking at your progress | Summary session, with summary of all previous sessions. | SIST/CBT | 7 |
| 8. Booster quiz | Quiz with 16 questions on all taught techniques. | SIST/CBT | 13 |

CBT, cognitive–behavioural therapy; SIST, social interaction skills training; YP, Young People.

problems. Adolescents that receive psychological care during participation in the study will not be allowed further participation.

## Materials

All outcome measures are completed at all three time points.

### Primary outcomes

*Social anxiety.* The Social Anxiety Scale for Adolescents (SAS-A)[18] will be used to assess social anxiety. The SAS-A contains 22 statements that are rated on a five-point Likert Scale (eg, I worry about what others say about me; I get nervous when I meet new people). A high score indicates high levels of social anxiety. The SAS-A contains three subscales, namely, fear of negative evaluation (FNE), social avoidance and distress in new situations (SAD-new) and social avoidance and distress in general (SAD-general). Cronbach's alphas are satisfactory for the subscales, being $\alpha=0.91$, 0.83 and 0.76, respectively.[18] Further research also yielded satisfactory Cronbach's alphas of $\alpha=0.89$, 0.80 and 0.70 for FNE, SAD-new and SAD-general respectively.[19]

*Body esteem.* The subscale 'appearance' from the Body Esteem Scale For Adolescents and Adults (BESAA)[20] is used to assess participant's attitudes and feelings about their appearance. A higher score indicates more satisfaction with one's appearance. This subscale consists of 10 statements, such as 'I like how I look in photos'. The items are rated on a five-point Likert Scale ranging from 'never' to 'always'. The subscale demonstrates excellent internal consistency in a sample of American girls ($\alpha=0.92$) and boys ($\alpha=0.90$)[21] and in a sample of Dutch university students ($\alpha=0.90$).[22]

### Secondary outcomes

*Aspects of self-worth.* The Dutch version of the Self-Perception Profile for Adolescents,[23] the Competentiebelevingsschaal voor Adolescenten (CBSA),[24] will be used to assess different aspects of self-worth. The CBSA consists of eight specific domains and a global self-worth subscale. The specific domains are scholastic competence, social competence, athletic competence, physical appearance, job competence, romantic appeal, behavioural conduct and close friendship. The CBSA consists of 35 items, each consisting of two contrasting statements (eg, some teenagers are *not* happy with the way they look, BUT other teenagers *are* happy with the way they look). The adolescent is asked to pick the statement that is most representative of them and choose whether the statement is 'sort of true for me' or 'really true for me'. All domains demonstrate good to excellent internal reliabilities, ranging from $\alpha=0.74$ to $\alpha=0.93$. The global self-worth subscale also demonstrates excellent reliability, with $\alpha=0.80$–0.89.[23]

*Perceived stigmatisation.* The Perceived Stigmatisation Questionnaire (PSQ)[25] will be used to measure stigmatisation behaviours commonly experienced by people with a visible difference. The PSQ is divided into three subscales, namely, absence of friendly behaviour, confused/staring behaviour and hostile behaviour. These subscales amount to a total of 21 items concerning stigmatising behaviours, such as 'people avoid looking at me' and 'people call me names'. Participants are asked to rate how often they experience certain behaviour on a five-point Likert Scale. A high score reflects high perceived stigmatisation. The three subscales have been established,[26] and the PSQ has adequate reliability in a sample of children and adolescents with a visible difference ($\alpha=0.81$).[27]

*Health-related quality of life.* The EuroQol-5D-5L (EQ-5D-5L)[28] is used to assess health-related quality of life. The scale has five dimensions, namely, mobility, self-care, usual activities, pain/discomfort and anxiety/depression. Each dimension can be assessed on five levels, which range from no problems to extreme problems. The EQ-5D-5L has good absolute and relative discriminatory power and good convergent validity.[29]

*Life engagement.* The Life Engagement Scale[30] will be used to measure the extent that worries or feeling bad about the way you look stops you from engaging in life activities such as going to a social event or doing sports. The scale consists of ten questions on which the adolescent can rate how much their worries or feeling bad about the way they look stopped them from doing activities on a four-point scale. A high score indicates less life engagement. This instrument was purposively constructed for adolescents with a visible difference. The scale has good internal consistency, with $\alpha=0.93$ for girls and $\alpha=0.96$ for boys.[30]

*Appearance-related distress.* The Mirror, Mirror (Spiegeltje, Spiegeltje)[31] Questionnaire will be used to assess appearance-related distress. This questionnaire consists of 58 statements about appearance (eg, my appearance makes me insecure), with a five-point Likert Scale ranging from 'never' to 'always'. The questionnaire also contains a section with 38 body parts where adolescents have to indicate on a five-point Likert Scale how happy they are with said body part (eg, skin colour and the mouth). A high score indicates more dissatisfaction with appearance. As this questionnaire is new and unpublished, it has not been validated. We aim to validate this questionnaire using the data obtained in this study.

*Depressive symptoms.* The Child Depression Inventory-2 (CDI-2)[32] will be used to assess depressive symptoms. The questionnaire consists of 28 items with three answer options (eg, I am sad sometimes, I am often sad and I am always sad). The scale consists of the subscales emotional problems, divided into negative mood/physical symptoms and negative self-esteem, and functional problems divided into ineffectiveness and interpersonal problems. A high score indicates more depressive symptoms. The questionnaire has demonstrated excellent internal reliability both for 13–16 year olds ($\alpha=0.89$) and for 17–21 year olds ($\alpha=0.85$).[32] The scale has good test–retest reliability (r=0.60).[32]

## Recruitment

Participants are recruited through several channels. The first is through patient organisations. Patient organisations will be approached and asked if they are willing to advertise the Dutch YP Face IT Study on their website and in their newsletters. If an adolescent is willing to participate, he/she is encouraged to send an email to the research team, providing his/her contact details. The adolescent will then receive an information package containing a patient information letter, an informed consent form and a response card. The second channel

is through our website (www.faceitvoorjongeren.nl). On our website, adolescents can read information about YP Face IT and contact the research team. If they do so, they will receive an information package. The third channel is through the Erasmus MC-Sophia Children's Hospital. Subjects will be drawn from the department of child and adolescent psychiatry/psychology, the department of plastic and reconstructive surgery, the department of oral and maxillofacial surgery and special dental care and the department of dermatology. Subjects will be informed about the study by their treating doctor/clinician. It will be explicitly stated that participation is voluntary and that they can withdraw from the study at any time should they wish to do so. After informing the potential participant, they will receive an information package. If a potential participant does not have regular appointments at the Erasmus MC-Sophia Children's Hospital, we will send an information package to their home address.

A response card will be included in the information package. On this response card, adolescents can indicate whether they give permission to be contacted by the research team. If the response card has not been returned within 2 weeks, the research team will contact the adolescent.

If participants are willing to participate, they are asked to fill in a consent form. For participants younger than 16 years, both parents are required to sign the informed consent form. After receiving the informed consent form, the researchers will include the participant in the trial and send out the first set of online questionnaires.

Participants will receive a gift card worth €10, after completing the entire study. If a participant decides to withdraw from the study prematurely, the adolescent will not receive a gift card.

## Study design

A flow chart of the RCT is depicted in figure 1. Prior to baseline, the inclusion and exclusion criteria as mentioned before (with the exception of appearance-related distress) are queried in a telephone call with the parent or adolescent. Appearance-related distress is measured based on the questionnaires completed at T1 (baseline). Adolescents will receive an email with a link to all questionnaires, with the exception of the CBSA. They will receive this questionnaire by post. This procedure is the same for T2 (13 weeks, direct follow-up) and T3 (25 weeks, 3-month follow-up).

After completing the questionnaires, screening for appearance-related distress will take place by the research team. The scores on the SAS-A, BESAA and CDI-2 are considered. For the SAS-A and the BESAA, adolescents that score 0.5–2.0 SD above average will be included in the study. For the CDI-2, we will include adolescents scoring in the 70th–90th percentile, as the manual states that this corresponds to subclinical symptoms of depression.[32] Adolescents that score within the defined range for one or more of the questionnaires will be randomised into the study. Adolescents that score below this range will

van Dalen M, *et al. BMJ Open* 2021;**11**:e041449. doi:10.1136/bmjopen-2020-041449

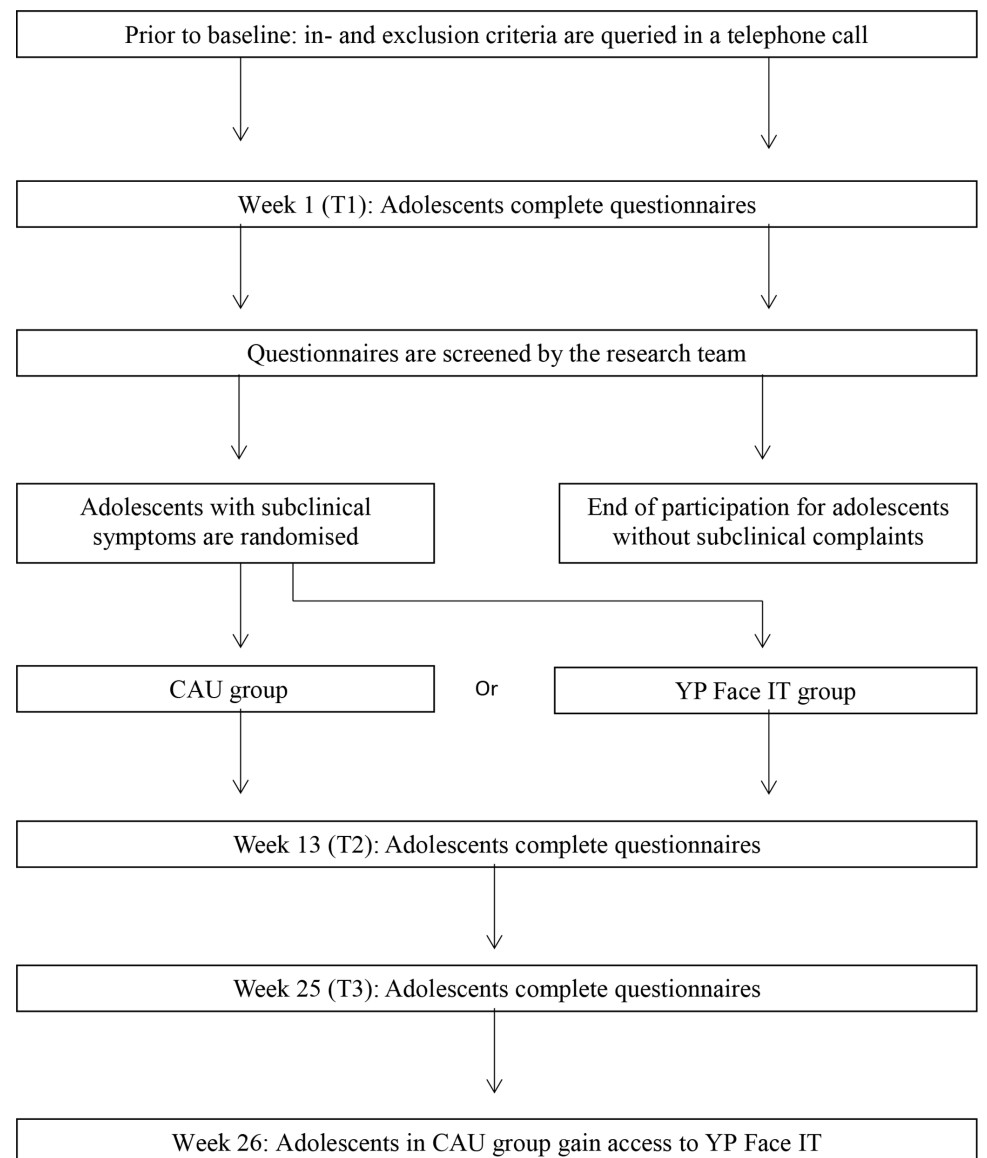

**Figure 1** . Flow chart of the study design. CAU, care as usual; YP, Young People.

be excluded from participation. Adolescents that score above this range will be contacted by the research team to assess whether there are clinical psychological symptoms. A phone call is made to both the adolescent and one of his/her parents. During this phone call, the Diagnostic and Statistical Manual of Mental Disorders (DSM) criteria for depressive disorder, social anxiety disorder or another suspected diagnosis will be asked. If suspicions of clinical psychological symptoms arise, the adolescents are referred to the appropriate psychological care. If no clinical symptoms are present, the adolescent is randomised into the study.

Adolescents with subclinical symptoms will be randomised to either CAU or YP Face IT. Adolescents in the CAU group will receive CAU and will complete T2 at 13 weeks and T2 at 25 weeks. The CAU group will have the opportunity to complete YP Face IT after their participation in the study. Adolescents in the YP Face IT group will receive CAU and YP Face IT and will complete

questionnaires immediately after completing YP Face IT (T2; 13 weeks) and 3 months after completing YP Face IT (T3; 25 weeks).

### Randomisation and blinding
Participants will be randomised to either the CAU or the YP Face IT group in a 1:1 ratio. Randomisation will be stratified by age (12–13, 14–15 or 16–17 years old). Randomisation will be done using a list with a random sequence, generated using a computer programme. With the stratification, six different types of blocks can be formed. Each block will independently vary between four and eight randomisation pairs. The adolescents are informed of the group they are randomised to through email. The researcher analysing the data will be blind to the randomisation and will not be involved in monitoring the adolescents in the intervention. This researcher also supervises data collection by students, who are not blinded. The participants, the clinical psychologist and

students monitoring the adolescents in the intervention will not be blind to the treatment conditions. When contacting the research team, the adolescents are explicitly instructed not to talk about the group they were allocated to.

If, through monitoring, the research team suspects that a participant experiences clinically significant symptoms, unblinding may take place. The participant will be excluded from the study and will be transferred to more appropriate care.

Participants that drop out after randomisation will be asked if they are willing to complete the questionnaires at T2 and T3. They are in no way obliged to do so.

### Sample size

For the sample size calculation, we applied a mixed-model Analysis of Variance (ANOVA) procedure, with a power of 0.80, a two-sided alpha of $\alpha=0.025$ (Bonferroni corrected for two primary outcome measures), a correlation between the repeated measurements of $r=0.70$ and three repeated linearly decreasing measurements. Bessell and colleagues[17] reported effect sizes of $d=0.83$ for appearance and $d=0.80$ for anxiety in a study on Face IT. However, as we are not certain that we will obtain such a large effect, we will consider a medium effect of $d=0.50$. To detect a medium effect, 56 participants are needed in both groups. As dropout is typically high in eHealth interventions,[33 34] we anticipated a dropout of 50%. Therefore, 224 participants are needed in total.

### Statistical analysis

The data will be analysed using multilevel linear regression analyses. There will be two levels in the models. The participants constitute the upper level and their repeated measures the lower level. For each outcome variable, a model will be postulated using treatment group, time and interaction with treatment group as fixed effects. Time will be entered in two ways: categorical and continuous. First, for categorical, we will postulate a model with T2 (direct follow-up) and T3 (3-month follow-up) using T1 (baseline) as reference group. Second, for continuous, we will apply the logarithm of time, as generally treatments have the largest effect at start and the level of functioning stabilises on the long run. The deviance statistic using restricted maximum likelihood will be applied to determine the covariance structure. The deviance test will be used to determine whether the more parsimonious (log) model is a too rigorous oversimplification.[35] Effect sizes will be calculated from dividing differences between the primary time point (ie, 3 months) estimations and baseline by the estimated baseline SD. Analyses are done on an intention-to-treat base.

Participants who do not experience subclinical symptoms at T1, and are thus not randomised, will not be included in the analyses.

### Monitoring

As the risks associated with this study are minimal, monitoring will take place once a year. An independent investigator will randomly check study documents, participant selection and participant safety. As minimal risks are involved, no interim analyses will be done.

Auditing may be done by the Erasmus MC-Sophia Children's Hospital.

### Patient and public involvement (PPI)

The original YP Face IT intervention was developed in collaboration with adolescents and the charity Changing Faces.[16] For the Dutch translation, six adolescents and their parents have read and reviewed the translations. Prior to this RCT, an acceptability and feasibility study was conducted. In this study, 14 adolescents completed YP Face IT and participated in an interview. Feedback was asked on the intervention, as well as on the general study procedures (ie, information packages, questionnaires and contact with the research team). Patient associations will be invited to help develop our dissemination strategy.

### Trial status

The trial described in this paper has started in September 2019. Recruitment of participants will be until November 2020. Data collection will be until May 2021, after which the data will be analysed.

## ETHICS AND DISSEMINATION
### Ethics, consent and permission

This study will be conducted in accordance to the Declaration of Helsinki.

Research ethics approval was obtained from the medical ethics review committee in Rotterdam (reference number MEC-2018-052/NL63955.078.18). Any amendments to the study protocol and (serious) adverse events will be reported to this committee.

### Data management and confidentiality

To ensure anonymous processing of the data, every participant will receive a study number. All research data will be processed using only this study number. The handling of data will comply with the European General Data Protection Regulation. The study master file and all research data will be saved for 15 years, unless participants or their parents/guardians do not give consent for saving their data.

All data will be stored in locked cabinets or password-protected documents. The data, including the final dataset, can be accessed only by members of the research team.

### Dissemination of research findings

Results of this study will be presented on national and international conferences and in peer-reviewed scientific journals, within the scope of the target groups. The results will also be relayed to patients through patient organisations. Participants are also given the option to sign up for a brief study report after the results are finalised. If deemed effective, YP Face IT will be implemented in hospitals in the Netherlands.

## DISCUSSION

The Dutch YP Face IT Study aims to test the Dutch version of YP Face IT. With this RCT, we aim to retrieve data on the effectiveness of the intervention.

### Strengths and limitations

A particular strength of this study is that it follows a stage model of behavioural therapies research[36] and has excellent methodological quality. Previous research has lacked methodological rigour, due to small sample sizes, not including a control group or not blinding personnel or participants.[15] Due to the nature of this study, blinding of participants is not possible. However, this study does include an adequate sample size and includes a control group and blinding of relevant personnel. With this methodological quality, we hope to be able to provide evidence base for YP Face IT and thus provide one of the first evidence-based treatments for adolescents with a visible difference.[15] Furthermore, by recruiting our participants through several channels, we will be able to obtain a sample that is representative for Dutch adolescents in general rather than solely Rotterdam, where the research team is located. Because the intervention is available online, participants throughout the entire country are able to participate without adding extra burden to participating.

In addition to strengths, this study also has some limitations. First, the intervention is available for a range of conditions causing a visible difference. The overall group results might not be generalisable to one specific group. Second, participants are not blind to the treatment condition. Participants are informed about the randomisation groups. They are also informed that, if randomised to the CAU group, they will have the possibility to access YP Face IT after the study. This is a more ethical option than not offering treatment after the study, and this knowledge may help keep attrition low. However, waiting-list designs may lead to an overestimate of the treatment effect.[37] Furthermore, this study includes only self-report measures. Diagnostic interviews or parent reports could provide more enriched data. However, due to time and financial constraints, these data will not be collected.

**Author affiliations**
¹Department of Child and Adolescent Psychiatry/Psychology, Erasmus MC-Sophia Children's Hospital, Rotterdam, Netherlands
²Department of Dermatology, Centre of Pediatric Dermatology, Erasmus MC-Sophia Children's Hospital, Rotterdam, Netherlands
³Department of Plastic and Reconstructive and Hand Surgery, The Dutch Craniofacial Centre, Erasmus MC Sophia-Children's Hospital, Rotterdam, Netherlands
⁴Department of Oral and Maxillofacial Surgery, The Dutch Craniofacial Centre, Erasmus MC-Sophia Children's Hospital, Rotterdam, Netherlands
⁵Department of Psychiatry, unit of Medical Psychology and Psychotherapy, Erasmus MC, Rotterdam, Netherlands
⁶Department of Health and Social Sciences, University of the West of England, Bristol, UK
⁷Research Institute of Child Development and Education, University of Amsterdam, Amsterdam, The Netherlands
⁸Academic Center for Child Psychiatry Levvel/Department of Child and Adolescent Psychiatry, Amsterdam University Medical Center, Amsterdam, The Netherlands

**Contributors** JO, HW and EMWJU developed the study design. RT made the statistical analysis plan for the study. JO initiated the study and obtained funding. SGMAP, M-LA, IM, MK, RT, MH and EMWJU reviewed and supported the funding application. JO and EMWJU supervised the study and data collection. MvD conducted the data collection, along with psychology and medical students. MvD wrote the first draft of the manuscript. All authors read and approved the final manuscript.

**Funding** This work was supported by a grant from the Innovatiefonds Zorgverzekeraars [grant number B 17–133/Dossier 3446]. The funding body has no role in the design of the study and the collection, analysis and interpretation of the data during this study.

**Competing interests** None declared.

**Patient and public involvement** Patients and/or the public were involved in the design or conduct or reporting or dissemination plans of this research. Refer to the Methods and analysis section for further details.

**Patient consent for publication** Not required.

**Provenance and peer review** Not commissioned; externally peer reviewed.

**Data availability statement** No data are available. Data sharing is not applicable to this article as no new data were created or analysed in this manuscript.

**ORCID iDs**
Marije van Dalen http://orcid.org/0000-0001-8361-8604
Jolanda Okkerse http://orcid.org/0000-0002-7644-2548

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
