## [Reviewer comments · BMJ Open]

ARTICLE DETAILS

TITLE (PROVISIONAL)	The Dutch YP Face IT Study, investigating online psychological treatment for Adolescents with a Visible Difference: Study Protocol of a Randomised Controlled Trial
AUTHORS	van Dalen, Marije; Pasmans, Suzanne G. M. A.; Aendekerk, Marie-Louise; Mathijssen, Irene; Koudstaal, Maarten; Timman, Reinier; Williamson, Heidi; Hillegers, Manon; Utens, Elisabeth; Okkerse, Jolanda

VERSION 1 – REVIEW

REVIEWER	Bethan Davies NIHR Mindtech Medtech Cooperative, University of Nottingham, UK
REVIEW RETURNED	19-Aug-2020

GENERAL COMMENTS	Thank you for submitting your protocol “The Dutch YP Face IT Study, investigating online psychological treatment for Adolescents with a Visible Difference: Study Protocol of a Randomised Controlled Trial” for consideration in BMJ Open. It’s great to see cross-country collaboration in trying out interventions in another country. This is a decent protocol – but I do have some comments relating to its presentation. Abstract: this is good, but I would suggest some rewording to make it clear that this is an two-armed RCT. Language: I’m not quite sure the word “suffering” should be used. In my background (health psychology), we tend to say “person experiencing xyz” rather than “suffering” as that can be quite subjective. Introduction: Good background to what visible difference is and its psychological impact. Could it be clarified that YP Face IT hasn’t been subject to an RCT in its home country (UK)? It seems a bit unusual to me that the intervention hasn’t yet undergone an RCT in its country of origin before being evaluated in another country. Some further explanation might be beneficial here. Methods and Analysis – some more information about the YP Face IT intervention would be beneficial. E.g. is it entirely self-led, or is there remote ‘therapist’/‘coach’ guidance – and what does it consist of? Do parents have a role in it as well? Anticipated dropout is 50% - where did you gather this estimate from?
---

	Some spelling typos in places e.g. “Until the July 2020” – don’t think ‘the’ is needed. The outcome measures are well described. Could you mention the three time points in the text (not just in Figure) and outline how they are collected at each point? I wasn’t sure if it was paper or online etc. Also I wasn’t sure whether the person collecting outcome data and analysis will be blinded to condition. In terms of patient and public involvement – have patients been involved in developing the participant facing materials for the RCT? Is adherence to the intervention assessed, e.g. through analytics? And is there a ‘minimum’ of the intervention that must be completed to be considered a ‘completer’? In terms of analysis – will intent-to-treat analyses be done, or ‘completers only’ – or both? Will there be any qualitative data collected alongside quantitative data?
--	---

REVIEWER	Hsing-Yi Chang National Health Research Institutes
REVIEW RETURNED	14-Oct-2020

GENERAL COMMENTS	This protocol was designed to test the effectiveness of a program helping teenagers with visible difference. The intervention was an online psychotherapy. Stratified randomization was proposed. Measurements would be taken at three time points. Investigators proposed to use multilevel linear regression from repeated measures. 1. Sample size was calculated accounting for repeated measurements. Was stratification considered? 2. The multilevel regression is a reasonable analysis to use. Why logarithm of time was considered? Could they not be treated as categorical data?
---

VERSION 1 – AUTHOR RESPONSE

Reviewer Comments to Author:

Reviewer: 1

Reviewer Name: Bethan Davies

Institution and Country: NIHR Mindtech Medtech Cooperative, University of Nottingham, UK Please state any competing interests or state ‘None declared’: None

Comments to the Author

1. Thank you for submitting your protocol “The Dutch YP Face IT Study, investigating online psychological treatment for Adolescents with a Visible Difference: Study Protocol of a Randomised Controlled Trial” for consideration in BMJ Open. It’s great to see cross-country collaboration in trying out interventions in another country. This is a decent protocol – but I do have some comments relating to its presentation.

We thank Dr. Davies for her conscientious review and helpful feedback. We have provided our answers to the comments below.

2. Abstract: this is good, but I would suggest some rewording to make it clear that this is an two-armed RCT.

We thank Dr. Davies for this suggestion. We have rephrased the abstract in lines 51-52 to emphasise that this is a two-armed RCT: “In this two-armed Randomised Controlled Trial, 224 adolescents will be randomised to care as usual or YP Face IT.”

3. Language: I’m not quite sure the word “suffering” should be used. In my background (health psychology), we tend to say “person experiencing xyz” rather than “suffering” as that can be quite subjective.

In hindsight, we agree with Dr. Davies that suffering might not be the correct term to use. We have replaced “suffering” with “experiencing” throughout the entire paper.

4. Introduction: Good background to what visible difference is and its psychological impact. Could it be clarified that YP Face IT hasn’t been subject to an RCT in its home country (UK)? It seems a bit unusual to me that the intervention hasn’t yet undergone an RCT in its country of origin before being evaluated in another country. Some further explanation might be beneficial here.

We thank Dr. Davies for her compliments on the introduction. As outlined in the introduction there are very little interventions available for adolescents with a visible difference. To our knowledge, no Dutch intervention is available. In the UK, The version for adults, Face IT had already been tested in an RCT and was found to be effective (see Bessel et al, 2012). Furthermore, qualitative evidence indicates that YP Face IT is probably effective for adolescents (see Williamson, Griffiths & Harcourt, 2015). Based on these positive findings we first conducted our acceptability and feasibility study in The Netherlands (see point 9). Thereafter we decided to start an RCT in The Netherlands.

The main reason why an RCT has not been conducted in the UK is lack of financial funding. A grant for an RCT in The Netherlands was awarded, so an RCT was started in The Netherlands with support of the developer of YP Face IT, Heidi Williamson.

5. Methods and Analysis – some more information about the YP Face IT intervention would be beneficial. E.g. is it entirely self-led, or is there remote ‘therapist’/‘coach’ guidance – and what does it consist of? Do parents have a role in it as well?

We thank Dr. Davies for these useful questions. We have added an extra paragraph that hopefully answers these questions: “YP Face IT is designed as a self-help intervention, without any active guidance by a psychologist. Adolescents work through the sessions on their own. Parents receive a short description of each session and will receive the reminders, but are not involved in the sessions themselves.” (lines 155-158)

We have also included an extra table, Table 1, that outlines the content for each session of the intervention.

6. Anticipated dropout is 50% - where did you gather this estimate from?

Our estimate was based on previous literature showing drop-out rates to be generally high in eHealth interventions. We apologise for not having included this literature. We have rephrased our sample size calculations to include relevant literature: “As drop-out is typically high in eHealth interventions (see 33, 34), we anticipated a drop-out of 50%. Therefore, 224 participants are needed in total.” (lines 331-332).

7. Some spelling typos in places e.g. “Until the July 2020” – don’t think ‘the’ is needed.

We apologise for the typos. We have checked the paper carefully and removed the typos we found.

8. The outcome measures are well described. Could you mention the three time points in the text (not

just in Figure) and outline how they are collected at each point? I wasn't sure if it was paper or online etc. Also I wasn't sure whether the person collecting outcome data and analysis will be blinded to condition.

We thank Dr. Davies for her compliments. We have mentioned the three time points, along with the way the data was collected, in the text: "Appearance-related distress is measured based on the questionnaires completed at T1 (baseline). Adolescents will receive an e-mail with a link to all questionnaires, with the exception of the CBSA. They will receive this questionnaire by post. This procedure is the same for T2 (13 weeks, direct follow-up) and T3 (25 weeks, three months follow-up)." (lines 277-281)

We have also included information on whether the person involved in collecting outcome data and analysis is blinded: "The researcher analysing the data will be blind to the randomisation and will not be involved in monitoring the adolescents in the intervention. This researcher also supervises data collection by students, who are not blinded. The participants, the clinical psychologist and students monitoring the adolescents in the intervention will not be blind to the treatment conditions." (lines 311-315)

9. In terms of patient and public involvement – have patients been involved in developing the participant facing materials for the RCT?

We thank Dr. Davies for this question. Prior to this RCT an acceptability and feasibility study was conducted (see <https://www.trialregister.nl/trial/7138>). In this trial 14 adolescents completed YP Face IT and participated in an interview with the current first author, Marije van Dalen. Our main goal was to assess the acceptability and feasibility of YP Face IT, but the interview also focused on the general study procedures and participant facing materials. We have added this in the manuscript on lines 358-361: "Prior to this RCT an acceptability and feasibility study was conducted. In this study 14 adolescents completed YP Face IT and participated in an interview. Feedback was asked on the intervention, as well as on the general study procedures (i.e. information packages, questionnaires and contact with the research team)."

10. Is adherence to the intervention assessed, e.g. through analytics? And is there a 'minimum' of the intervention that must be completed to be considered a 'completer'?

As described in lines 159-163 monitoring of the intervention takes place. We will register when a participant does a session and how long each session takes. Time will then be used as a categorical and continuous variable in the final analyses.

The intervention website automatically registers how long a participant takes to do a session. However, we found that this is quite unreliable. If someone leaves his/her browser open, the session will continue to be registered. In the acceptability and feasibility study some participants were registered as having taken more than a day to do a session. Hence, we decided not to use this metric. We will analyse based on an intention-to-treat. Hence, we will not be considering people completers or non-completers. We will analyse everyone that has done at least one session of the program.

11. In terms of analysis – will intent-to-treat analyses be done, or 'completers only' – or both?

Our analysis will be done on an intention-to-treat base. We have added this to lines 346-347.

12. Will there be any qualitative data collected alongside quantitative data?

As described in our answer to question 9, we have conducted an acceptability and feasibility trial. The results of these interviews are being analysed and will be reported in a scientific journal in due time.

Reviewer: 2

Reviewer Name: Hsing-Yi Chang

Institution and Country: National Health Research Institutes Please state any competing interests or state 'None declared': None.

Comments to the Author

This protocol was designed to test the effectiveness of a program helping teenagers with visible difference. The intervention was an online psychotherapy. Stratified randomization was proposed. Measurements would be taken at three time points. Investigators proposed to use multilevel linear regression for repeated measures.

1. Sample size was calculated accounting for repeated measurements. Was stratification considered? We have stratified in three age groups, but we did not take this into account in the sample size calculation. The sample size calculation is based on an expected effect of $d=0.50$, and meant to detect an overall effect. We did not mean to study changes in the separate stratified groups.

2. The multilevel regression is a reasonable analysis to use. Why logarithm of time was considered? Could they not be treated as categorical data?

We agree with Dr. Chang's suggestion. As stated in lines 340-342 the logarithm of time was considered because interventions typically have the most effect directly after they are administered. It is likely that this effect will lessen over time. By using the logarithm of time, we intent to capture this effect. By using time as a logarithmic effect we can fit a more parsimonious model. But we agree that this may be an oversimplification. Thus we will first postulate a model with two categorical time effects (T2 and T3, T1 as reference) and then a model with time as a logarithmic effect. We will determine whether the more parsimonious (log) model is a too rigorous simplification with the deviance test (Singer & Willett, 2003).

We have changed this in lines 336-344: "For each outcome variable a model will be postulated using treatment group, time and interaction with treatment group as fixed effects. Time will be entered in two ways: categorical and continuous. First, for categorical we will postulate a model with T2 (direct follow-up) and T3 (three month follow-up) using T1(baseline) as reference group. Second, for continuous we will apply the logarithm of time, as generally treatments have the largest effect at start and the level of functioning stabilizes on the long run. The deviance statistic using restricted maximum likelihood will be applied to determine the covariance structure. The deviance test will be used to determine whether the more parsimonious (log) model is a too rigorous oversimplification (35)."

VERSION 2 – REVIEW

REVIEWER	Bethan Davies NIHR Mindtech MedTech Cooperative, University of Nottingham
REVIEW RETURNED	04-Jan-2021
GENERAL COMMENTS	Thank you to the authors for showing how each reviewer comment was addressed. I am satisfied with the adjustments/revisions that have been made to the manuscript. Many thanks.